# Handwritten Amharic Character Recognition System Using Convolutional Neural Network

## Abstract

Amharic language is an official language of the federal government of the Federal Democratic Republic of Ethiopia. Accordingly, there is a bulk of handwritten Amharic documents available in libraries, information centres, museums, and offices. Digitization of these documents enables to harness already available language technologies to local information needs and developments. Converting these documents will have a lot of advantages including (i) to preserve and transfer history of the country (ii) to save storage space (ii) proper handling of documents (iv) enhance retrieval of information through internet and other applications. Handwritten Amharic character recognition system becomes a challenging task due to inconsistency of a writer, variability in writing styles of different writers, relatively large number of characters of the script, high interclass similarity, structural complexity and degradation of documents due to different reasons. In order to recognize handwritten Amharic character a novel method based on deep neural networks is used which has recently shown exceptional performance in various pattern recognition and machine learning applications, but has not been endeavoured for Ethiopic script. The CNN model is trained and tested our database that contains 132,500 datasets of handwritten Amharic characters. Common machine learning methods usually apply a combination of feature extractor and trainable classifier. The use of CNN leads to significant improvements across different machine-learning classification algorithms. Our proposed CNN model is giving an accuracy of 91.83% on training data and 90.47% on validation data.

## 1 Introduction

In recent years, there is much interest in the area of handwritten documents recognition. Between the handwritten documents and printed documents, automatic handwritten document recognition is more challenging. Handwritten characters written by different persons are not identical and vary in both size and shape. Numerous variations in writing styles of individual character make the recognition task difficult. The similarities in distinct character shapes, the overlaps, and the interconnections of the neighbouring characters further complicate the problem. Recognition is an area that covers various fields such as, face recognition, finger print recognition, image recognition, character recognition, numerals recognition, etc. Handwritten Character Recognition System Sarkhel et al. (2016) is an intelligent system able to classify handwritten Characters as human see. There have been different methods that are used for offline handwritten document recognition. In the conventional methods with features engineered manually and using different classification algorithms to classify the characters based on the extracted features. On the other hand, deep learning algorithms such as convolutional neural networks are able to do the feature extraction by themselves from the raw images of the handwritten document and classify characters on those features learned. Deep learning methods show better performance in different researchers work for handwritten recognition task. Among the deep learning methods convolutional neural networks, which is the one proposed for this research work, are the most commonly used algorithms. Deep Neural networks consist of input layer and multiple nonlinear hidden layers and output layer, so the number of connections and trainable parameters are very large. The deep neural network needs very large set of examples to prevent over fitting. One class type of Deep Neural Network with comparatively smaller set of parameters and easier to train is Convolutional Neural Network (CNN) Liang et al.

(2016). CNN is a multi-layer feed-forward neural network that extracts features and properties from the input data (images or sounds). CNN is trained with neural network back-propagation algorithm. CNN have the ability to learn from high-dimensional complex inputs, nonlinear mappings from very large number of data (images or sounds) Maitra et al. (2015). The advantage of CNN is that it automatically extracts the salient features which are invariant and a certain degree to shift and shape distortions of the input characters Shin et al. (2016).Another major advantage of CNN is the use of shared weight in convolutional layers, which means that the same filter is used for each input in the layer. The share weight reduces parameter number and improves performance Bai et al. (2015a). Recently, Convolutional Neural Network (CNN) Lecun & Bengio (1995) is found efficient for handwritten character recognition due to its distinct features. CNNs add the new dimension for image classification systems and recognizing visual patterns directly from pixel images with minimal pre-processing. In addition, CNN automatically provides some degree of translation invariance. A CNN based model was tested on UNIPEN Guyon et al. (1994) English character dataset and found recognition rates of 93.7 percent and 90.2 percent for lowercase and uppercase characters, respectivelyYuan et al. (2012). Amharic language has its own alphabet which is significantly different from other alphabets such as Latin alphabet. Although the Ethiopic alphabet called Fidel has recently been standardized to have 435 characters. However the most commonly Ethiopic script used by Amharic has 265 characters including 27 labialized characters (characters representing two sounds) ጊ ሟ ሯ ሲ ሺ ቋ ቄ ቢ ቲ ቿ ኋ ኗ ኟ ካ ኰ ዚ ዢ ዲ ጇ ጓ ጪ ጫ ጲ ፋ ጐ ጕ ኧ and 34 base characters with six orders representing derived vocal sounds of the base character, 21 symbols for numerals and 8 punctuation marks. There is no capital and lower case distinction. When we see the features of Amharic characters they have the following basic characteristics: Each symbol is written according to the sound it have when pronounced. Vowels are created by modifying the base characters in some form. The symbols are written in disconnected manner e.g ሀ, ሁ, ሂ, ሃ. The direction of writing the script is from left to right and top to bottom sequence. There is a proportional space between characters and words. The Amharic language alphabet is conveniently written in a tabular form of seven columns as shown in Table 1 where each column corresponds to vocal sounds in the order of ä, u, i, a, e, ə, and o. Several handwritten scriptures and documents written in this language are available on paper or on any other material. Converting the handwritten documents into digital forms helps us to process, share and store them in electronic form. The conventional way of converting the handwritten Amharic documents in to an electronic form, which is done by typing on the keyboard, is very time consuming, error prone and tedious. Due to the keyboard layout for Amharic characters which takes an average of two keystrokes to write one Amharic character, the conventional way of converting handwritten Amharic documents will be very difficult. This emphasizes the need for an automatic handwritten Amharic character recognition system which converts handwritten texts into machine–readable code that can be accepted by a computer for further processing. Amharic handwriting recognition is challenging due to mainly two reasons. First, it has huge number of symbols compared to that of the alphabet system. Second, most characters are very similar in shape. This is because of the minimal modification performed to get order of a character. For instance, ቡ comes from ቢ, ሰ comes from ሱ , and ሁ comes from ሀ . Most core characters also show similarities to one another. One common feature is a mark of palatalization which sets off palatal ሸ from ሰ, ቸ from ተ, ጀ from ደ, ኘ from ነ and so on. The interclass variability is even minimal in case of handwritings where mostly these modifications are forgotten or placed at a wrong position. Lack of standard way of writing aggravates the problem by increasing the intra-class variability. Nowadays, it is becoming increasingly important to have information in digital format for increased efficiency in data storage, retrieval and to make them available for users. Although a lot of work and research has been done for handwritten character recognition for other languages like English and Asian languages such as Japanese, Chinese and Korean, there is only a few research attempts at Amharic language. A few works has been reported in scientific literature related to the recognition of Amharic printed and handwritten document recognition. In Assabie & Bigün (2011) they develop a recognition system for Ethiopic script using

direction field tensor mechanism. Their system is developed by extracting primitive structural features and their spatial relationship. Since there is no standard database for Ethiopic text they use thirty pages scanned image from newspaper, books and clean printouts. The achieved performance for their system is 87%. They did not consider handwritten documents in their dataset. In Birhanu & Sethuraman (2015) they have used ANN approach for recognition of real life documents. They collected their dataset from 'Addis Zemen' newspaper, Amharic Bible, 'Federal NegaritGazeta' newspaper and the fiction 'FikerEskeMekabir'. The performance of their system for a new test set is 11.40% which is not satisfactory and the proposed system is trained with printed documents rather than handwritten. In Meshesha & Jawahar (2007) they develop a system which uses a principal component and linear discriminant analysis followed by a decision directed acyclic graph based support vector machine based classifier. Existing methods including those discussed above for Amharic document recognition systems, employ manually designed feature extractor and learned classifier and most of them use printed documents rather than handwritten. It is not easy to design an optimal feature extractor for a particular application. Hence, the performance of these algorithms is not satisfactory. Various methods have been proposed and high recognition rates are reported for the handwritten recognition of other languages. In Bai et al. (2015b) they propose shared-hidden-layer deep convolutional neural network (SHL-CNN) for image character recognition. In SHL-CNN, the hidden layers are made common across characters from different languages, performing a universal feature extraction process that aims at learning common character traits existed in different languages such as strokes, while the final softmax layer is made language dependent, trained based on characters from the destination language only. The effectiveness of the learned SHL-CNN is verified on both English and Chinese image character recognition tasks, showing that the SHL-CNN can reduce recognition errors by 16-30% relatively compared with models trained by characters of only one language using conventional CNN, and by 35.7% relatively compared with state-of-the-art methods. A modified LeNet-5 which is one of common CNN model with special settings of the number of neurons in each layer and the connecting way between some layers is proposed by Yuan et al. (2012) for offline handwritten English character recognition. They used the UNIPEN lowercase and uppercase dataset in their experiments and attain a recognition rate of 93.7% and 90.2% for uppercase and lowercase respectively. Authors in Wu et al. (2014) proposed handwritten recognition method for Chinese character based on relaxation convolutional neural network (R-CNN) and alternately trained relaxation convolutional neural network. The relaxation convolutional layer in their model, unlike the traditional convolution layer, does not require neurons within a feature map to share the same convolutional kernel, endowing the neural network with more expressive power. Authors in Zhong et al. (2015a) applied multi-pooling and data augmentation with non-linear transformation to a convolutional neural network (CNN) for multi-font printed Chinese character recognition (PCCR). They propose a multi-pooling layer on top of the final convolutional layer; this approach is found to be robust to spatial layout variations and deformations in multi-font printed Chinese characters. Outstanding recognition rate of 94.38% is achieved by combining the multi-pooling and data augmentation techniques and 99.74% by applying the multi-pooling and data augmentation techniques with non-linear transformation jointly. In Yang et al. (2015) the authors proposed an enhancement of deep convolutional neural network for recognition of online handwritten Chinese character. The enhancement is done by incorporating a variety of domain specific knowledge, including deformation, non-linear normalization, imaginary strokes, path signature and 8-directional features. The contribution in this work is twofold. First the domain specific technologies are investigated and integrated with the deep convolutional neural network to form a composite network to achieve improved performance. Second, the resulting deep convolutional neural networks with diversity in their domain knowledge are combined using a hybrid serial-parallel strategy. A promising accuracy of 97.20% and 96.87% are achieved using CASIA-OLHWDB1.0 and CASIA-OLHWDB1.1 dataset for Chinese character respectively. In He et al. (2015) an effective method to analyze the recognition confidence of handwritten Chinese character based on softmax regression score of a high performance convolutional network is studied. In Zhong et al. (2015b)

authors proposed a deeper architecture of CNN algorithm by using streamlined version of GoogLeNet. They used the ICDAR 2013 offline Chinese character recognition system competition dataset. With incorporation of traditional directional feature maps the proposed GoogLeNet models achieve an accuracy of 96.35% and 96.74% as single and ensemble models respectively. A handwritten Hangul character recognition system using deep convolutional neural network by proposing several novel techniques to increase the performance and training speed of the networks is done by Kim & Xie (2015). In Anil et al. (2015) Malayalam handwritten character recognition using the convolutional neural network is developed and in their work they discussed the CNN is better than the conventional handcrafted feature extractor based systems. Deep learning based large scale handwritten Devanagari character recognition is proposed by Acharya et al. (2015) with focus on the use of dropout and dataset increment approach to increase the test accuracy of their deep learning model. A combination of four different pattern analysis techniques are used to develop a powerful and efficient system for handwritten Telgu character recognition system is proposed by Soman et al. (2013). Their system embodies convolutional neural networks, principal component analysis, support vector machines and multi-classifier systems. As compared to the handwritten automated character recognition system discussed above the Amharic character recognition system is the least studied subject both in the conventional handcrafted feature extractor based systems as well as deep learning based convolutional neural networks. In the proposed method, both the feature extraction and classification tasks are done through learning from labelled data. This method overcomes the problems faced by the existing methods. Visual recognition system using convolutional neural networks Lecun et al. (1998) have shown a significant improvement in recent years. Record-breaking results have been obtained using these methods. This has motivated the researcher to investigate the success of the CNN algorithms on this challenging problem. Visual recognition using convolutional neural networks enables us to train the complete system from end to end.

## 2 Amharic Handwritten Character Recognition System

### 2.1 General Architecture

The task of the recognition is done using convolution neural networks. The proposed system consists of two main components: pre-processing and segmentation unit, feature extraction and classification. Digitalization, noise removal, binarization, normalization belongs to the preprocessing step in the proposed system. In the segmentation step different segmentation methods are used such as line segmentation, word segmentation and character segmentation from the given scanned image of handwritten document to extract individual characters. The two essential components in recognition, feature extraction and classification, both are done in our CNN model.

### 2.2 Architecture of Proposed System

In a convolutional neural network the input to a convolutional layer is M x M x C image where M is the height and width of the image, M X M is number of pixels in image and C is number of channels per pixel. For gray scale image have one channel C = 1 but RGB image have three channels C = 3. A CNN consists of a number of layers (convolutional layers, pooling layers, fully connected layers). The convolutional layer will have K filters (kernels) of N x N x R where N is height and width of filter (kernels) and R is the same number of image channels C or less and may vary for each filter (kernel). The filter (kernel) convolved with the image to produce K feature maps of size M-N+1. Each feature map is then pooled typically with mean or max pooling over q x q where q is range between 2 to 5 for large inputs. After the convolutional layers and pooling layers there may be any number of fully connected layers as in a standard multi-layer neural network. Convolutional layer acts as a feature extractor that extracts salient features of the inputs such as corners, edges, endpoints. Then, the convolutional layer applies its activation function to add non

linearity to its output. The activation function that is used in our proposed approach is Rectified Linear Units (ReLU). The ReLU non-linearity used as an activation function to the output of every convolutional layer and fully connected layer. The ReLU Nair & Hinton (2010) increases the nonlinear properties of the decision function and of the overall network without affecting the receptive fields of the convolution layer Elsawy et al. (2017).

After each convolutional layer, there may be a pooling layer. The pooling layer takes small rectangular blocks from the convolutional layer and subsamples it to produce a single output from that block. The pooling layer reduces the resolution of the image that reduces the precision of the translation (shift and distortion) effect. There are several ways to do this pooling, such as taking the average or the maximum, or a learned linear combination of the neurons in the block. Our pooling layers will always be maxpooling layers; that take the maximum of the block that they are pooling. All the max-pooling is carried out over a 2 x 2 pixel window [23].

Finally, after several convolutional and pooling layers, the high-level reasoning in the neural network is done via fully connected layers. A fully connected layer takes all neurons in the previous layer (be it fully connected, pooling, or convolutional) and connects it to every single neuron it has. Fully connected layers are not spatially located anymore (you can visualize them as one-dimensional), so there can be no convolutional layers after a fully connected layer. Figure 1 shown the proposed CNN architecture for Amharic handwritten character recognition that describe as follow: INPUT-->CONV-->RELU-->CONV-->RELU-->MAX POOL-->CONV-->RELU-->MAX POOL-->CONV-->RELU-->MAX POOL-->CONV-->RELU-->MAX POOL--> FC-->RELU--> FC-->Softmax.

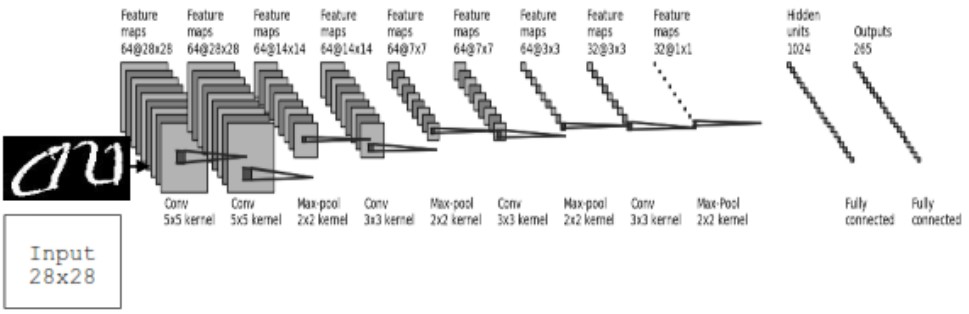

Figure 1: The proposed CNN architecture for Amharic handwritten character recognition.

The first and the second convolution layers have 64 filters of size 5x5, the next two conv layers have 64 filters with size 3x3 and the last conv layer has 32 filters with kernel size of 3x3. The second important layer in convolutional neural networks is the pooling layer. This layer simply acts as a downsampling filter. After the second conv layer a max-pooling with 2x2 pixel window is used. After this polling layer we have 3 conv---- > max pool layers with 3x3 kernel size for the convolution layer and 2x2 pixel window for the pooling layer. The size of the convolution filters are initialized based on experimental results. Experimentally select filter sizes for each conv layer by seeing the performance improvement of the network during training and validation. After each convolution layer the RELU activation function is used. The rectifier activation function is used to add non linearity to the network. The Flatten layer is use to convert the final feature map, which is an output from the last max pooling layer, into a one single 1D vector. This flattening step is needed so that you can make use of fully connected layers after some convolutional/maxpool layers. In the end the features in two fully-connected (Dense) layers which is just artificial neural networks (ANN) classifier is used. In the last fully connected layer (Dense(265, activation = "softmax")) the net outputs distribution of probability of each

class. Dropout is a regularization method, where a proportion of nodes in the layer are randomly ignored (setting their weights to zero) for each training sample. This drops randomly a proportion of the network and forces the network to learn features in a distributed way. This technique also improves generalization and reduces the problem of overfitting. A dropout of 0.5 at the last fully connected layer is used. The RMSprop optimizer to update filter weights and improve algorithm functionality is used. The default setting of this optimizer since it is recommended to use the default setting in the documentation is used. Batch size of 70 and run the network for 70 epochs is used. The batch size and the number of epochs are selected based on the best performance measurement values of the proposed CNN network during experimental set up. Python programming language with keras deep learning library with theano backend on CPU was used during the development of the convolutional neural network architecture. Image processing libraries designed for python such as numpy, opencv2, skyimage used for processing the input image for the conv net.

## 2.3 Architecture of Proposed System

In research like character recognition for handwritten documents using deep neural networks, the primary task is collecting required data and preparing it for further processing. The Amharic character benchmark dataset is still not available for the research community in public and this was the major challenge during this research work. Since the research is dealing with pattern recognition, or more specifically character recognition, the data collected are of two types. The first being data for training the convolutional neural network (recognition engine), the second will be for testing the performance of the CNN model. The Amharic Handwritten Character dataset used in our system is created by collecting the variety of handwritten Amharic characters from hand writings of different individuals from diverse fields. 500 dataset for each character were collected and having (500x265) 132,500 datasets in total. The data was collected from 250 persons who write each character 2 times in a white paper. The selected individuals can write the Amharic alphabet randomly with different educational background and different age ranges. Each individual wrote each character on the forms shown in figure 2. We divided this dataset in two groups: 20 percent for validation/ testing and the remaining for training our system. Handwritten documents are then scanned with 332 dpi Samsung galaxy note 4 CamScanner mobile application and cropped for individual characters with a size of 28 x 28 pixels and pre-processed before feeding it to the CNN using semi-automated algorithm. The collected data was labelled based on the 265 classes for each character by using integer values 0 to 264 for each class of the character. The disadvantage of convolutional neural network is its rapacious appetite for labelled training data. Hence, real-world data collection and applying image pre-processing methods are needed to make the system more robust and efficient. A large handwritten character dataset have been collected in our work and labelled to train as well as to evaluate the system performance compared with other researcher works for Amharic character recognition system.

## 2.4 Challenges in Amharic Character Recognition

The challenges in handwritten character recognition vary among different languages due to distinct shape, strokes and numbers of characters. In Amharic script there are characters which have similar structure which differs with each other with a little curve, line and strokes and this becomes even more challenging since such structural variations are forgotten during handwriting. Some examples of two different characters written similarly are shown in figure 3. The other big challenge is unavailability of public datasets to use for such recognition systems which needs large amount of data to be trained.

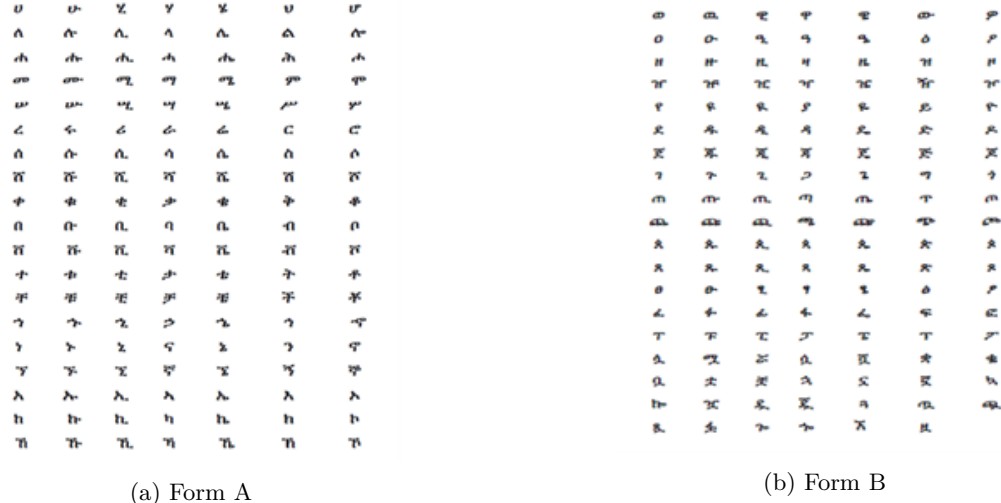

(a) Form A                    (b) Form B

Figure 2: Data collection forms for Amharic characters

Figure 3: Different characters written similarly.

## 3  Findings and Discussion

In this section, the performance of the CNN architecture was investigated for training and validating Amharic character recognition system. ConvNets have a large set of hyper-parameters and finding the perfect configuration for your problem domain is a challenge. Different configurations of the proposed network were explored and attempted to optimize the parameters based on the validation and training set accuracy. The performance of our recognition system was evaluated by using the collected dataset discussed earlier. From the collected dataset 20% (26,500) for validation and 80% (106,000) for training the proposed convolutional neural network architecture discussed earlier was used. The training and validation datasets are evenly distributed over the underlying 265 classes. In experimental set up data augmentation was used to increase the number of training dataset to avoid overfitting. The performance of our proposed system was checked with and without data augmentation and training with data augmentation shows better performance result. By changing different parameter values such as epoch size, batch size, optimizer selection, dropout and the layers of the network until the best fit model is found. In this section the results of experiments are presented. Based on experimental analysis, the CNN architecture discussed earlier is used for the proposed Amharic handwritten character recognition system. We perform different conv, max pool and FC layer configuration and observe the training and validation accuracy for a given CNN layer configuration to select the best depth of the proposed CNN. We selected the CNN layer configuration with the best performance result for our training and validation dataset. Then we improve the performance of

the selected CNN architecture by tuning other parameters of the network. The other network parameter we change it during our experimental analysis was the batch size selection. Table 1 illustrates an experiment conducted to observe the effect of batch size on the performance of the proposed CNN model with varying the batch size and using RMSprop optimizer with learning rate of 0.001. Using this experiment a batch size of 70 selected which gives better accuracy. For the other experiments we use 70 for batch size and tune other parameters to get improved performance.

Table 1: Accuracy with varying batch size using RMSprop optimizer and 20 epochs

| Batch size | Training acc. | Validation acc. | Training loss | Validation loss |
|---|---|---|---|---|
| 60 | 86.7% | 88.7% | 0.45 | 0.4 |
| 65 | 87.2% | 88.8% | 0.42 | 0.4 |
| 70 | 87.4% | 89.4% | 0.42 | 0.38 |
| 75 | 86% | 89% | 0.46 | 0.4 |
| 80 | 86% | 88% | 0.46 | 0.42 |
| 85 | 86% | 88.5% | 0.45 | 0.41 |
| 90 | 86% | 88.1% | 0.47 | 0.42 |

The other parameter which has an effect on the performance of our proposed model was the optimizer selection. Optimizers are used for weight update of the network and they have their own behaviors. For our proposed model the RMSprop optimizer gives better results compared with stochastic gradient decent and Adam optimizer. Increasing the epoch size from 20 to 30 we get better results with 89.99% for training accuracy and 90.19% for validation accuracy. The training loss reduces from 3.80 to 0.32 and the validation loss reduces from 1.96 to 0.36. With 40 epochs the training accuracy is 90.40% and validation accuracy of 90.37%. The training loss decreases from 3.80 to 0.31 and validation loss decreases from 1.96 to 0.35.Increasing the epoch size to 70 increases the training accuracy to 91.83% and the validation accuracy to 90.47%. We have get a little increase in the validation accuracy which means still the CNN model is improving its performance on test sets. The training loss decreases from 3.87 to 0.27 and the validation loss decreases from 1.87 to 0.36. Increasing the epoch size above 70 increases the processing time of the system. Due to this we take 70 as epoch size for our final model. Therefore we are using 70 both for epoch and batch size respectively.The graphs in figure 4 show the accuracy and loss of training and validation respectively with 70 epochs and 70 batch size.

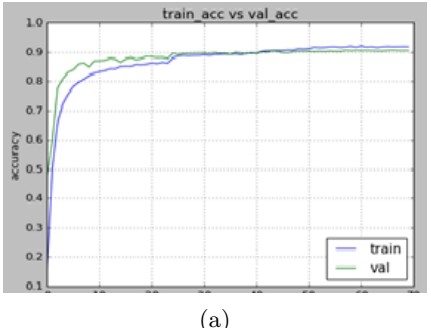
(a)

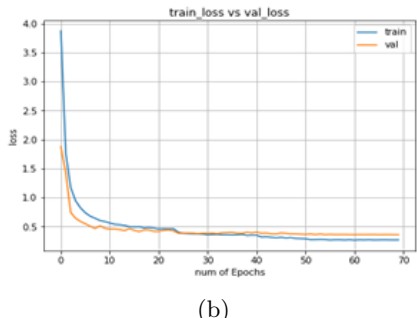
(b)

Figure 4: (a) validation vs training accuracy (b) validation vs training loss

## 4 Conclusion and Recommendation

In this research we use convolutional neural networks for recognition of Amharic handwritten characters with 265 character classes. It is well known that convolutional neural networks are the current state of the art algorithm for classifying image data.

We have collected large amount of Amharic characters from individual handwriting on prepared form for data collection. The collected handwritten document is scanned and pre-processed to get a 28 x 28 .jpg character images which are given as input to the convolutional neural network for classification. We presented a new public dataset for Amharic character dataset which is publicly available for any researcher and access it through contact address of the author. We developed a deep learning based Amharic handwritten character recognition system. To get the best fit model of CNN based architecture a lot of trial and error network configuration tuning mechanism has been used. Using the developed CNN model we have achieved better recognition accuracy compared with other research works based on conventional hand craft feature extraction based methods. We have achieved an accuracy of 91.83% on training dataset and 90.47% on validation dataset. Even though there are a lot of work for recognition of handwritten characters for English, Chinese, Arabic and some other Indian languages only little work is done for Amharic language. The previous research works did not use deep learning methods for Amharic character recognition. Due to lack of research works on the area there is a big challenge to get dataset for Amharic language. In this research we develop a dataset which can be used by other researchers in the future. By increasing the size of the collected dataset and proposing new deep learning algorithms we will increase the performance of the system. In the future, we want to extend this algorithm for recognition of Amharic words by creating new datasets.

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
