# OpenReview forum: "Handwritten Amharic Character Recognition System Using Convolutional Neural Networks"
_ICLR.cc/2020/Conference — Reject_

### Official Review · AnonReviewer1 · 2019-10-22
**Official Blind Review #1**

**Rating:** 1

**Review:**

This paper try to use CNN to build recognizer for handwritten Amharic characters. The CNN they used is simple and standard. Apparently this paper no novelty at all. They just apply CNN to a new task. This kind of work is not qualified for ICLR at all.

There are also some problems in paper organization. They should split the introduction part into several paragraphs to improve reading experience. And it seems that they forget to add reference part.

Given the quality of the writing and content, I decide to reject this paper.

**Experience Assessment:**

I have published one or two papers in this area.

**Review Assessment: Checking Correctness Of Derivations And Theory:**

N/A

**Review Assessment: Checking Correctness Of Experiments:**

I did not assess the experiments.

**Review Assessment: Thoroughness In Paper Reading:**

N/A

---

### Official Review · AnonReviewer2 · 2019-10-23
**Official Blind Review #2**

**Rating:** 1

**Review:**

In this paper, the authors collect and preprocess a large amount of handwritten Amharic characters and train a deep convolutional network to successfully perform character recognition on a subset of the data.

I vote to reject this paper. The formatting of the paper needs work, and the work is not substantially novel.

For some reason, many sections of the paper were ill-formatted, perhaps due to using an insufficiently portable submission format. In addition, though the paper cited a few references throughout, the final reference list was not available. In addition, handwritten character recognition, by itself, is not a new field, and the authors did not contribute sufficiently to the underlying theory or mechanics. Characters with arguably the same level of complexity, such as Chinese characters, have been thoroughly explored. The application of existing technology to a different script does not, by itself, compel me to feel that this is an ICLR worthy paper.

In the future, the authors might wish to show how their particular architecture (or processes) might be better suited to their task than other architectures, or show how they have, in some way, improved the science of character recognition. That being said, I think it is quite notable the effort that went into creating the data set for Amharic character recognition. I would encourage the authors to distribute this data set so that others may join in the research efforts.

**Experience Assessment:**

I have published one or two papers in this area.

**Review Assessment: Checking Correctness Of Derivations And Theory:**

N/A

**Review Assessment: Checking Correctness Of Experiments:**

I assessed the sensibility of the experiments.

**Review Assessment: Thoroughness In Paper Reading:**

N/A

---

### Official Review · AnonReviewer4 · 2019-11-02
**Official Blind Review #4**

**Rating:** 1

**Review:**

This paper considers the problem of character recognition applied to handwritten Amharic text. The authors collect a dataset of handwritten Amharic characters and apply a CNN model for character recognition. The task is contextualized within an extensive description of related work both in Amharic and other languages. The dataset is novel and would be of interest to the character recognition community, and I would encourage the authors to present it in its own right along with technical details about its collection. Unfortunately, due to methodological issues (detailed below), I do not think that the machine learning results in this paper are ready for publication at ICLR. The machine learning techniques applied are of limited novelty, however, the dataset is certainly a novel contribution.

Methodological issues:

The paper describes dividing the dataset in an 80-20 train-validation split. To the best of my understanding, there is no held out test set, and so we cannot know the performance of the model on unseen data. It appears that validation accuracy was taken into account when selecting hyperparameters, and so, validation accuracy also does not represent the model's performance on unseen data.

I would recommend that the authors introduce a test set into their dataset split, or designate the validation part of the dataset as a test part use cross validation for hyperparameter tuning.

The authors present the train and validation numbers on their dataset, but it is difficult to know the impact of the result without comparing to a baseline of some kind. It is challenging to compare to prior work since, according to the authors, prior work on Amharic character recognition has been focused on printed text.  However, a simple non-neural baseline would be illuminating.

Recommendations:

1. The authors should report results on an unseen test dataset, rather than train and validation sets.
2. The authors spend a lot of time motivating the use of deep learning in the introduction, and later on describing convolutional neural networks, ReLU, fully connected layers, etc. in great details. I believe that this space in the paper could be better utilized describing the things that are unique to the work presented.
3. It is really important to be able to see (a) the references and (b) the example characters, as it helps readers to situate the work and to understand the specific challenges addressed. I urge the authors to prioritize these technical details in submissions of future versions of this work.
4. Please consider separating out the description of related work into its own section. While it is useful to describe state of the art systems applied to related datasets, it is not necessary to go into great technical detail, especially if the methods applied are quite different from those attempted in this paper.
5. Based on the challenging nature of Amharic character recognition, it would be extremely helpful to see F1 numbers or a confusion matrix. While Figure 3 shows that different characters can be written similarly, it would be great to provide a quantitative measure of this phenomenon.
6. For a figure such as Figure 3, please provide an indication (e.g. the character number from 0-265, or a printed version of the character) of what the drawn character is supposed to look like, to help readers who cannot read Amharic script.

Questions about dataset
1. Please clarify what you mean by "the data collected are of two types". Does this refer to the train/validation split?
2. It would be great to know more about how many individuals were selected (and the choices made about their demographics) for writing the example characters, and whether there were any interesting variations observed in the dataset based on attributes highlighted in the paper (e.g. age range)
3. Please explain the meaning of 'Form A' and 'Form B' in Figure 2.
4. In the text, features such as a mark of palatalization are noted. Are these addressed in the data collection? Please give further details about how the dataset addresses the special features of the Amharic script.
5. At the top of page 6, it says that the data was labelled. Could you say more about the annotators (multiple?), and report inter annotator agreement? Or, did the writers produce each character based on a specific prompt, such that the labels are known to be correct?
6. Section 3 mentions data augmentation. Could you describe this in more detail?
7. Are the images in Figure 3 from your dataset? Please show some examples from the new dataset you have collected!


**Experience Assessment:**

I do not know much about this area.

**Review Assessment: Checking Correctness Of Derivations And Theory:**

N/A

**Review Assessment: Checking Correctness Of Experiments:**

I assessed the sensibility of the experiments.

**Review Assessment: Thoroughness In Paper Reading:**

I read the paper thoroughly.

---

### Comment · AnonReviewer2 · 2019-10-12
**References missing?**

The paper seems to be missing the references, at least from my end. Did this get clipped somehow?

In addition, it seems like there are some character formatting issues. Particularly on page 4 and (perhaps?) page 2.

Is there any possible resolution to these issues?

---

> ### Author Response · Authors · 2019-10-28
> **References missing issue**
>
> I think the latex format was mistakenly used from my side and you are correct the references are missed. The Amharic characters are also not displayed in the latex format given. I will correct the comments in my final version.

---

### Decision · Program_Chairs · 2019-12-19

**Decision:**

Reject

**Comment:**

The submission proposes to use CNN for Amharic Character Recognition.   The authors used a straight forward application of CNNs to go from images of Amharic characters to the corresponding character.  There was no innovation on the CNN side. The main contribution of the work is the Amharic handwriting dataset and the experiments that were performed.

The reviewers indicated the following concerns:
1. There was no innovation to the method (a straight forward CNN is used) and is likely not of interest to the ICLR community
2. The dataset was divided into train/val split and does not contain a held-out test set.  Thus it was impossible to determine the generalization of the model.
3. The paper is poorly written with the initial version having major formatting issues and missing references. The revised version has fixed some of the formatting issues.  The paper still need to having more paragraph breaks to help with the readability of the paper (for instance, the introduction is still one big long paragraph).  The terminology and writing can also be improved.  For instance, in section 2.3, the authors write that "500 dataset for each character were collected".  It would be clearer to say that "500 images for each character were collected".

The submission received low reviews overall (3 rejects), which was unchanged after the rebuttal.  Due to the general consensus, there was limited discussion.  There were also major formatting issues with the initial submission.  The revised version was improved to have proper inclusion of Amharic characters in the text, missing figures, and references.  However, even after the revision, the paper still had the above issues with methodology (as noted by R4) and is likely of low interest for the ICLR community.

The Amharic handwriting data and experiments using a CNN can be of interest to the different community and I would recommend the authors work on improving their paper based on reviewer comments and submit to different venue (such as a workshop focused on character recognition for different languages).